# Efficient Reasoning via Thought Compression for Language-Guided Segmentation

## Abstract

Chain-of-thought (CoT) reasoning has significantly improved the performance of large multimodal models in language-guided segmentation, yet its prohibitive computational cost, stemming from generating verbose rationales, limits its real-world applicability. We introduce WISE (Wisdom from Internal Self-Exploration), a novel paradigm for efficient reasoning guided by the principle of *thinking twice—once for learning, once for speed*. WISE trains a model to generate a structured sequence: a concise rationale, the final answer, and then a detailed explanation. By placing the concise rationale first, our method leverages autoregressive conditioning to enforce that the concise rationale acts as a sufficient summary for generating the detailed explanation. This structure is reinforced by a self-distillation objective that jointly rewards semantic fidelity and conciseness, compelling the model to internalize its detailed reasoning into a compact form. At inference, the detailed explanation is omitted. To address the resulting conditional distribution shift, our inference strategy, WISE-S, employs a simple prompting technique that injects a brevity-focused instruction into the user's query. This final adjustment facilitates the robust activation of the learned concise policy, unlocking the full benefits of our framework. Extensive experiments show that WISE-S achieves state-of-the-art zero-shot performance on the ReasonSeg benchmark with 58.3 cIoU, while reducing the average reasoning length by over **5×**—from 112 to just 23 tokens.

## 1 Introduction

Language-guided segmentation, the task of localizing an object within an image based on a natural language description, is a cornerstone of modern vision-language research. While early successes focused on simple referring expressions like "the car" (Yu et al., 2016; Kazemzadeh et al., 2014), the field's frontier has rapidly advanced towards *reasoning segmentation* (Lai et al., 2024). This more challenging task requires models to interpret complex, multi-step commands that are often compositional, relational, or rely on commonsense knowledge—for instance, "segment the car that is partially obscured by the bus." Successfully tackling such prompts necessitates a shift from

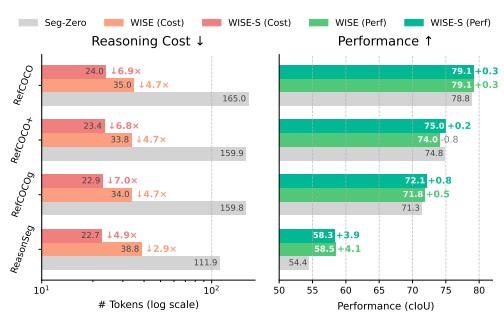

Figure 1: WISE balances cost and performance by separating reasoning for training and inference.

basic recognition to a multi-step cognitive process, where a model must deconstruct the query into a coherent chain of visual search and logical validation steps.

Recent Large Multimodal Models (LMMs) (Shang et al., 2024; Bai et al., 2025), augmented with Chain-of-Thought (CoT) prompting strategies (Wei et al., 2022; Ma et al., 2025), have emerged as a powerful solution for this task. By generating an explicit, step-by-step textual rationale, these models can effectively reason through complex instructions to identify the correct target. However, this enhanced reasoning capability is intrinsically tied to the generation of long, verbose text sequences (Wang et al., 2025; Feng et al., 2025). This creates a critical performance-efficiency

paradox: the very mechanism that drives accuracy introduces prohibitive computational costs and high inference latency, severely limiting the applicability of these models in real-world scenarios like robotics or interactive assistants where efficiency is paramount. **The fundamental challenge, therefore, is to decouple a model's reasoning capability from this costly, verbose output.**

In this work, we address this challenge by proposing **WISE (Wisdom from Internal Self-Exploration)**, a novel training and inference paradigm designed for efficient reasoning via thought compression. Our framework is guided by the principle of *thinking twice—once for learning, once for speed*. Instead of a simple 'think → answer' sequence, we train the model to generate a structured, three-part response: a **concise rationale** ($R_c$), the **final answer** ($A$), and then a **detailed explanation** ($R_d$). By compelling the model to commit to the concise rationale upfront, we leverage the fundamental autoregressive nature of decoders. This structure ensures that the generation of the final answer and, critically, the detailed explanation are both conditioned on the initial concise thought, $P(A, R_d | I, T, R_c)$. This conditioning forces the concise rationale to become a potent and sufficient summary of the entire reasoning process, as a poor summary would make it impossible for the model to generate a coherent, logically consistent detailed explanation.

This unique training structure is reinforced by a self-distillation objective that explicitly rewards the semantic fidelity between the concise rationale and the detailed explanation, while penalizing the verbosity of the former. This process encourages the model to internalize its elaborate reasoning capabilities into a compact, efficient policy. To ensure this learned policy is robustly activated at inference—where the detailed explanation is entirely omitted to maximize speed—our WISE framework culminates in WISE-S, a simple, zero-overhead prompting strategy. This final adjustment injects a brevity-focused instruction into the user's query (as shown in the Fig. 1), mitigating the conditional distribution shift between training and inference and ensuring the model consistently defaults to its more efficient reasoning mode.

Our contributions are as follows:

- We introduce WISE, a novel end-to-end framework for efficient reasoning segmentation. Its unique training paradigm decouples the verbose reasoning required for robust learning from the concise rationale needed for fast inference.

- We propose a self-distillation mechanism, built upon a unique *concise → answer → detailed* generation sequence that leverages autoregressive conditioning to effectively train the model to compress its own reasoning capabilities.

- We demonstrate through extensive experiments that our full framework, WISE-S, achieves state-of-the-art zero-shot performance on the ReasonSeg benchmark with 58.3 cIoU, while simultaneously reducing the average reasoning length by over **5×**.

## 2 RELATED WORK

### 2.1 LANGUAGE-GUIDED SEGMENTATION

Language-guided segmentation aims to localize specific objects in an image based on natural language descriptions. Early work in this domain primarily focused on referring expression segmentation (Kazemzadeh et al., 2014; Yu et al., 2016), where queries are typically simple and direct, such as "the person in the red shirt". These foundational works established the core challenge of grounding linguistic phrases to pixel-level visual evidence.

The field has recently evolved towards the more challenging task of reasoning segmentation (Lai et al., 2024). This advanced task requires models to interpret complex, multi-step instructions that involve relational, spatial, or commonsense reasoning. Several recent works have explored using LMMs to tackle this challenge, bridging the gap between high-level reasoning and low-level segmentation (Lai et al., 2024; Chen et al., 2024; Ren et al., 2024). A key work that pushed the boundaries of this task is Seg-Zero (Liu et al., 2025b), which introduced a framework for learning reasoning capabilities "from zero". By employing pure reinforcement learning (RL) instead of supervised fine-tuning (SFT), Seg-Zero demonstrated that a model could learn to generate an explicit reasoning process and achieve strong zero-shot performance on complex benchmarks like ReasonSeg (Lai et al., 2024). Our work builds directly upon this paradigm, adopting the core idea of using a reason-

ing model to guide a segmentation model. However, we identify and address a critical limitation in these prior works: the high computational cost associated with their verbose reasoning chains.

## 2.2 Chain-of-Thought in Large Multimodal Models

Chain-of-Thought (CoT) prompting has emerged as a powerful technique for unlocking the complex reasoning abilities of Large Language Models (LLMs) (Wei et al., 2022; Zhang et al., 2022; Yu et al., 2023). By prompting a model to generate a sequence of intermediate reasoning steps before providing a final answer, CoT has significantly improved performance on a wide range of tasks, from arithmetic to symbolic reasoning. This principle has been successfully extended to LMMs, enabling them to deconstruct complex visual-linguistic queries into manageable steps (Chen et al., 2024; Shen et al., 2025; Tan et al., 2025).

In the context of reasoning segmentation, models like LISA (Lai et al., 2024) and Seg-Zero (Liu et al., 2025b) implicitly leverage a CoT-like process by generating a textual rationale to guide their final segmentation decision. The success of these methods demonstrates the power of explicit reasoning for complex visual grounding. While highly effective, this approach creates a performance-efficiency paradox: the model's accuracy is directly coupled with the generation of long, verbose textual outputs, leading to high latency and computational overhead. This trade-off presents a major obstacle for real-world applications. Our work directly confronts this challenge by seeking to decouple reasoning capability from the generation of costly, verbose text, a problem not explicitly addressed by prior CoT-based segmentation methods.

## 2.3 Knowledge Distillation and Thought Compression

Knowledge distillation is a common technique for model compression, where a smaller "student" model is trained to mimic the output of a larger, more powerful "teacher" model. This concept has been adapted to distill complex reasoning processes from large models, often by training a smaller model on the rationales generated by a teacher (Lightman et al., 2023; Uesato et al., 2022). Our work introduces a novel form of *self-distillation* specifically designed for compressing the CoT process within a single model.

Unlike traditional distillation, which relies on a separate teacher model, WISE trains the model to be its own teacher. During training, the model generates both a concise rationale ($R_c$) and a detailed explanation ($R_d$). The core innovation lies in our training objective, which simultaneously rewards the semantic fidelity between the detailed "teacher" explanation and the concise "student" rationale, while penalizing the verbosity of the latter. The structured generation sequence ($R_c \rightarrow A \rightarrow R_d$) and autoregressive conditioning are critical mechanisms that enable this effective self-distillation. This approach fundamentally differs from prior methods like Seg-Zero, which learn to generate a single, often verbose, rationale through reinforcement learning (Guo et al., 2025) but lack an explicit mechanism for compression. To our knowledge, WISE is the first framework that trains a model to explicitly internalize and compress its own reasoning process for efficient yet powerful reasoning segmentation.

## 3 Methodology

Our objective is to enhance the efficiency of reasoning segmentation models by compressing their Chain-of-Thought (CoT) process, without sacrificing performance. As shown in 2, we introduce **WISE**, a new training and inference paradigm that redefines the learning objective for a policy-based reasoning model, $\mathcal{F}_{\text{reason}}$. This model is trained using the GRPO reinforcement learning algorithm (Shao et al., 2024) and is decoupled from the downstream segmentation model, $\mathcal{F}_{\text{seg}}$, which remains frozen throughout the process.

## 3.1 Problem Formulation

Given an input image $I$ and a textual instruction $T$, our goal is to train a policy $\pi_\theta$, embodied by the reasoning model $\mathcal{F}_{\text{reason}}$, to generate an optimal set of geometric prompts $A$ (e.g., bounding boxes and points) that localize the target object. The baseline approach trains the policy to generate a sequence containing a detailed reasoning chain $\tau_d$ and the prompts $A$. The optimization objective

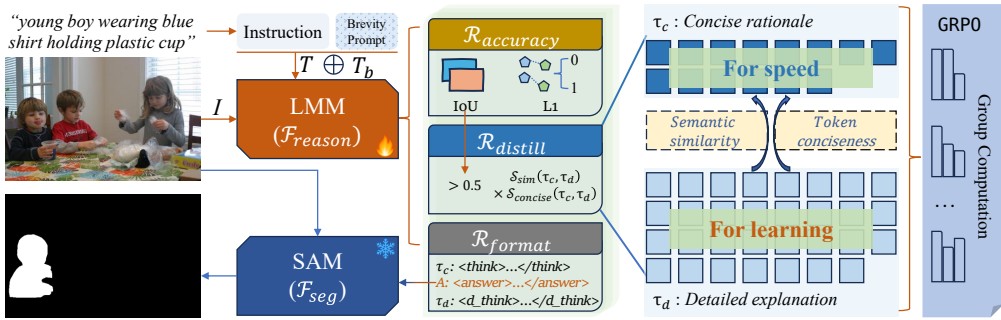

Figure 2: **Overview of the WISE training and inference framework.** During training (orange path), the reasoning model $\mathcal{F}_{\text{reason}}$ (LMM) takes an image $I$ and an instruction $T$ to produce a concise rationale ($\tau_c$), an answer with geometric prompts ($A$), and a detailed explanation ($\tau_d$). The process is optimized by the GRPO algorithm using a hierarchical reward signal composed of $\mathcal{R}_{\text{accuracy}}$ (IoU and L1 of prompts), $\mathcal{R}_{\text{format}}$, and a conditional self-distillation reward $\mathcal{R}_{\text{distill}}$. $\mathcal{R}_{\text{distill}}$ encourages $\tau_c$ to be semantically similar to $\tau_d$ but much shorter. The segmentation model $\mathcal{F}_{\text{seg}}$ (SAM) remains frozen and is not part of the training loop. During inference (blue path), the detailed explanation $\tau_d$ is omitted, and an optional brevity prompt ($T_b$) is used to ensure the generation of a highly compressed rationale for maximum efficiency.

is to find parameters $\theta^*$ that maximize the expected reward, which is calculated directly on the generated prompts:

$$\theta^* = \arg\max_\theta \mathbb{E}_{\pi_\theta}[\mathcal{R}_{\text{task}}(A, A_{gt})] \tag{1}$$

where $A_{gt}$ represents the ground-truth geometric prompts. For final evaluation, the generated prompts $A$ are fed into a fixed, pre-trained segmentation model $M = \mathcal{F}_{\text{seg}}(A)$, but $\mathcal{F}_{\text{seg}}$ and the final mask $M$ are not involved in the training loop of $\pi_\theta$. The primary motivation for our work is the high computational cost of generating the verbose $\tau_d$ at inference time.

### 3.2 THE WISE TRAINING PARADIGM

The core of WISE is to train the policy $\pi_\theta$ to learn a compressed reasoning representation. This is achieved by restructuring the generation sequence as a structured action space and introducing a hierarchical, conditional reward objective.

#### 3.2.1 STRUCTURED ACTION SPACE AND AUTOREGRESSIVE CONDITIONING

Instead of the standard action space $\mathcal{A}_{\text{baseline}} = (\tau_d, A)$, we define a new action space for training: $\mathcal{A}_{\text{train}} = (\tau_c, A, \tau_d)$, where $\tau_c$ is a concise rationale (`<think>`), $A$ is the answer (`<answer>`), and $\tau_d$ is a detailed explanation (`<d_think>`). The autoregressive policy $\pi_\theta$ models the joint probability of this sequence as:

$$\pi_\theta(\tau_c, A, \tau_d | I, T) = \pi_\theta(\tau_c | I, T) \cdot \pi_\theta(A | I, T, \tau_c) \cdot \pi_\theta(\tau_d | I, T, \tau_c, A) \tag{2}$$

This generation order imposes a powerful structural prior, compelling $\tau_c$ to act as a sufficient statistic for the generation of $\tau_d$, thereby creating ideal conditions for self-distillation.

#### 3.2.2 HIERARCHICAL REWARD OBJECTIVE

The learning process is guided by a hierarchical reward function. The base task reward, $\mathcal{R}_{\text{task}}$, ensures syntactic and geometric fidelity. It is a composite function composed of a format reward, $\mathcal{R}_{\text{format}}$, and an accuracy reward, $\mathcal{R}_{\text{accuracy}}$:

$$\mathcal{R}_{\text{task}}(\tau_c, A) = \mathcal{R}_{\text{format}}(\tau_c, A) + \mathcal{R}_{\text{accuracy}}(A, A_{gt}) \tag{3}$$

Following prior work (Liu et al., 2025b), $\mathcal{R}_{\text{format}}$ is a binary reward verifying the presence of required tags and a valid JSON structure. $\mathcal{R}_{\text{accuracy}}$ is the sum of binary rewards for the Intersection over Union (IoU) and L1 distance of the geometric prompts in $A$.

Building upon this baseline, the total training reward, $\mathcal{R}_{\text{train}}$, conditionally incorporates our novel self-distillation reward, $\mathcal{R}_{\text{distill}}$:

$$\mathcal{R}_{\text{train}} = \mathcal{R}_{\text{task}} + \mathcal{R}_{\text{distill}}(\tau_c, \tau_d) \cdot \mathbb{I}(\text{IoU}(A) > 0.5) \quad (4)$$

where $\mathbb{I}(\cdot)$ is the indicator function. The distillation reward $\mathcal{R}_{\text{distill}}$ is defined as the product of a similarity score and a conciseness score:

$$\mathcal{R}_{\text{distill}} = \mathcal{S}_{\text{sim}}(\tau_c, \tau_d) \cdot \mathcal{S}_{\text{concise}}(\tau_c, \tau_d) \quad (5)$$

The semantic similarity score, $\mathcal{S}_{\text{sim}}$, is computed as the cosine similarity between the sentence embeddings of the two rationales, obtained from a pretrained SentenceTransformer model $\mathcal{F}_{\text{ST}}$:

$$\mathcal{S}_{\text{sim}}(\tau_c, \tau_d) = \frac{\mathcal{F}_{\text{ST}}(\tau_c) \cdot \mathcal{F}_{\text{ST}}(\tau_d)}{\|\mathcal{F}_{\text{ST}}(\tau_c)\|\|\mathcal{F}_{\text{ST}}(\tau_d)\|} \quad (6)$$

The conciseness score, $\mathcal{S}_{\text{concise}}$, measures the fractional length reduction of the concise rationale relative to the detailed one, given by:

$$\mathcal{S}_{\text{concise}}(\tau_c, \tau_d) = \max\left(0, 1 - \frac{\text{len}(\tau_c)}{\text{len}(\tau_d)}\right) \quad (7)$$

where $\text{len}(\cdot)$ denotes the number of tokens. This conditional objective ensures the model only learns to distill reasoning pathways that are proven to be effective.

### 3.3 THE WISE-S INFERENCE STRATEGY

At inference time, the action space is reduced to $\mathcal{A}_{\text{infer}} = (\tau_c, A)$ by omitting the generation of $\tau_d$. This creates a conditional distribution shift, as the policy is now queried for $P(\tau_c|I, T; \text{goal} = A)$ instead of the training-time distribution that anticipated generating $\tau_d$. To mitigate this, the WISE-S strategy modifies the input instruction $T$ with a brevity-focused prompt $T_b = $ *'the shorter the better'*, creating a new inference prompt $T_S = T \oplus T_b$, where $\oplus$ denotes injection. The objective is to align the inference-time policy with the characteristics learned during training, specifically to promote the generation of a concise rationale ($\tau_c$) that is as brief as its training-time counterpart. This is reflected in the following approximation:

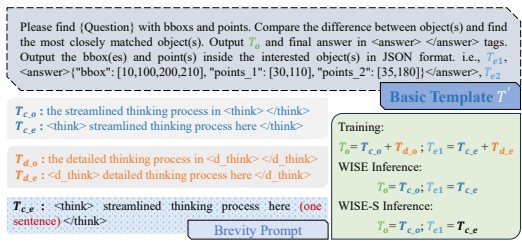

Figure 3: **Prompting Strategy for WISE.** The figure shows the basic instruction template and the specific components for generating concise ($T_c$) and detailed ($T_d$) rationales. Different combinations of these components are used for the training phase, standard WISE inference, and the brevity-focused WISE-S inference.

$$\pi_\theta(\tau_c, A|I, T_S) \approx \pi_\theta(\tau_c, A|I, T; \text{trained with } \mathcal{R}_{\text{distill}}) \quad (8)$$

This final step ensures the model robustly produces the highly compressed rationales learned during the WISE training phase.

## 4 EXPERIMENTS

**Datasets and Metrics.** To empirically validate our framework, we largely adhere to the evaluation protocols established by prior work (Liu et al., 2025b). We train our models on a 2,000-sample subset of the RefCOCOg dataset (Yu et al., 2016). Crucially, no human-annotated rationales are used during training, meaning the reasoning capabilities are learned from scratch. For evaluation, we assess out-of-domain reasoning capability on the challenging ReasonSeg benchmark (Lai et al., 2024) and test in-domain performance on the standard suites of RefCOCO, RefCOCO+, and Ref-COCOg (Yu et al., 2016). Segmentation accuracy is measured primarily by cumulative IoU (cIoU), while reasoning efficiency is quantified by the average number of generated reasoning tokens.

**Baselines.** Our primary baseline is Seg-Zero, which represents a strong, reasoning-augmented segmentation model. To demonstrate that WISE's effectiveness stems from its unique training structure rather than just a preference for shorter outputs, we implement two additional strong baselines (Aggarwal & Welleck, 2025) that directly optimize for brevity via reward shaping:

- **L1-Exact:** This baseline adds a reward term that is inversely proportional to the L1 distance between the generated rationale's length and a fixed, short target length.
- **L1-Max:** This baseline introduces a penalty for any rationale exceeding a predefined maximum length, encouraging the model to stay within a token budget.

**Implementation Details.** Our model architecture comprises a Qwen2.5-VL-7B (Bai et al., 2025) reasoning model ($\mathcal{F}_{\text{reason}}$) and a frozen SAM2-Large (Ravi et al., 2024) segmentation model ($\mathcal{F}_{\text{seg}}$). We optimize the reasoning model using the GRPO reinforcement learning algorithm (Shao et al., 2024) with the DeepSpeed library (Rasley et al., 2020). Training is conducted with a total batch size of 16, a learning rate of $1 \times 10^{-6}$, and the AdamW optimizer with a weight decay of 0.01. The user instruction for training and inference is shown in Figure 3.

**Inference Strategies.** We evaluate two variants of our trained model at inference time:

- **WISE (default):** This is our standard inference setting used across all experiments unless otherwise specified. It simply involves removing the instruction to generate the detailed explanation (<d_think>) from the training-time prompt, relying on the model's naturally learned concise policy.
- **WISE-S (shortened):** To fully and robustly activate the learned brevity, this variant incorporates an additional brevity-focused prompt, such as "*one sentence*", into the user instruction.

Table 1: Comparison with state-of-the-art methods on the **Zero-shot ReasonSeg** benchmark. WISE-7B and WISE-7B-S significantly outperform prior methods. *Re-evaluate with official checkpoints. †Reported in the original paper.

Table 2: Comparison with state-of-the-art methods on the **Referring Expression Segmentation** benchmark. Our models achieve competitive or superior performance against specialized methods while being more general.

| Method | ReasonSeg | | | |
|---|---|---|---|---|
| | val | | test | |
| | gIoU | cIoU | gIoU | cIoU |
| OVSeg (Liang et al., 2023) | 28.5 | 18.6 | 26.1 | 20.8 |
| ReLA (Liu et al., 2023) | 22.4 | 19.9 | 21.3 | 22.0 |
| LISA-7B (Lai et al., 2024) | 53.6 | 52.3 | 48.7 | 48.8 |
| SAM4MLLM (Chen et al., 2024) | 46.7 | 48.1 | - | - |
| Qwen2.5VL-3B + SAM2 | 53.8 | 44.1 | 47.6 | 37.4 |
| Seg-Zero-7B† (Liu et al., 2025a) | 62.6 | 62.0 | 57.5 | 52.0 |
| Seg-Zero-7B* (Liu et al., 2025a) | 60.9 | 57.4 | 57.7 | 54.4 |
| **WISE-7B (ours)** | **63.5** | **59.2** | **60.3** | **58.5** |
| **WISE-7B-S (ours)** | **63.5** | 58.8 | **60.3** | 58.3 |

| Method | RCO | RCO+ | RCOg |
|---|---|---|---|
| | testA | testA | test |
| LAVT (Yang et al., 2022) | 75.8 | 68.4 | 62.1 |
| ReLA (Liu et al., 2023) | 76.5 | 71.0 | 66.0 |
| LISA-7B (Lai et al., 2024) | 76.5 | 67.4 | 68.5 |
| PixelLM-7B (Ren et al., 2024) | 76.5 | 71.7 | 70.5 |
| MagNet (Chng et al., 2024) | 78.3 | 73.6 | 69.3 |
| PerceptionGPT (Pi et al., 2024) | 78.6 | 73.9 | 71.7 |
| Seg-Zero-7B† (Liu et al., 2025a) | 79.3 | 73.7 | 71.5 |
| Seg-Zero-7B* (Liu et al., 2025a) | 78.8 | 74.8 | 71.3 |
| **WISE-7B (ours )** | **79.1** | 74.0 | 71.8 |
| **WISE-7B-S (ours )** | **79.1** | **75.0** | **72.1** |

## 4.1 MAIN RESULTS

We conduct a series of experiments to validate the effectiveness and efficiency of our proposed WISE framework. We first present the main results, comparing WISE with state-of-the-art methods on both reasoning and referring segmentation benchmarks. We then conduct extensive ablation studies to deconstruct the key components of our method and analyze their individual contributions.

**State-of-the-Art on Reasoning Segmentation.** As shown in Table 1, on the challenging zero-shot ReasonSeg benchmark, our WISE models significantly outperform existing state-of-the-art methods. Specifically, our default WISE-7B model achieves 60.3 gIoU and 58.5 cIoU on the test set, surpassing the strong LISA-13B baseline by a large margin (+6.5 gIoU, +7.7 cIoU). This demonstrates the strong reasoning capability cultivated by our training paradigm. The shortened variant, WISE-7B-S, maintains this high level of performance, confirming that the concise rationales preserve the essential logical steps required for complex reasoning.

**Competitive Performance on Referring Segmentation.** Table 2 shows the performance on referring expression segmentation benchmarks. Despite being trained on a small subset of RefCOCOg,

Table 3: Performance and Efficiency Comparison on Referring and Reasoning Segmentation Benchmarks. Our WISE models drastically reduce token overhead (#Tok) while improving or maintaining accuracy (cIoU) across all benchmarks compared to the Seg-Zero and brevity-focused reward shaping methods (L1-Exact, L1-Max (Aggarwal & Welleck, 2025)). The sub-rows show the token reduction factor ($\times \downarrow$) and the absolute performance change ($\Delta$) against the Seg-Zero.

| Method | RefCOCO$_{testA}$ | | RefCOCO+ $_{testA}$ | | RefCOCOg $_{test}$ | | ReasonSeg (cIoU) | | | |
|---|---|---|---|---|---|---|---|---|---|---|
| | #Tok ↓ | cIoU ↑ | #Tok ↓ | cIoU ↑ | #Tok ↓ | cIoU ↑ | #Tok ↓ | val ↑ | #Tok ↓ | test ↑ |
| Seg-Zero | 165.1 | 78.8 | 159.9 | 74.8 | 159.8 | 71.3 | 117.5 | 57.4 | 111.9 | 54.4 |
| Seg-Zero+L1-Exact | 31.2 | 78.3 | 31.1 | 74.9 | 30.7 | 71.7 | 36.6 | 57.2 | 35.6 | 55.0 |
| Seg-Zero+L1-Max | **11.7** | 78.9 | **11.7** | 74.6 | **10.6** | 70.7 | **10.5** | 43.8 | **12.0** | 49.7 |
| **WISE** (ours) | 35.0 | **79.1** | 33.8 | 74.0 | 34.0 | 71.8 | 38.8 | **59.2** | 38.8 | **58.5** |
| *vs. Seg-Zero* | *(4.7× ↓)* | *(+0.3)* | *(4.7× ↓)* | *(-0.8)* | *(4.7× ↓)* | *(+0.5)* | *(3.0× ↓)* | *(+1.8)* | *(2.9× ↓)* | *(+4.1)* |
| **WISE-S** (ours) | 24.0 | 79.1 | 23.4 | **75.0** | 22.9 | **72.1** | 24.6 | 58.8 | 22.7 | 58.3 |
| *vs. Seg-Zero* | *(6.9× ↓)* | *(+0.3)* | *(6.8× ↓)* | *(+0.2)* | *(7.0× ↓)* | *(+0.8)* | *(4.8× ↓)* | *(+1.4)* | *(4.9× ↓)* | *(+3.9)* |

our models exhibit strong generalization. WISE-7B-S achieves competitive or even superior results compared to specialized methods, particularly on RefCOCO+ and RefCOCOg, highlighting its robustness and wide applicability without sacrificing performance on simpler, in-domain tasks.

**Breaking the Efficiency-Performance Trade-off.** Table 3 provides a comprehensive comparison of performance versus efficiency. Across all four benchmarks, both WISE and WISE-S drastically reduce the number of generated tokens compared to the Seg-Zero baseline. WISE-S, in particular, achieves a remarkable **4.9×** to **7.0×** reduction in reasoning length. More importantly, this massive gain in efficiency is not achieved at the cost of accuracy. In most cases, WISE-S *improves* performance, especially on the difficult ReasonSeg task (+3.9 cIoU). This result empirically validates our core thesis: by teaching a model to reason efficiently, we can break the conventional trade-off and achieve both speed and effectiveness simultaneously.

## 4.2 ABLATION STUDIES

To understand the source of WISE's effectiveness, we conduct a series of detailed ablation studies. All ablations are performed using the 7B model variant.

### 4.2.1 DECONSTRUCTING THE THOUGHT COMPRESSION MECHANISM

We first analyze the two core components of our proposed thought compression: the generation order and the self-distillation reward.

**Generation Order is Crucial.** Table 4 investigates the impact of the generation sequence. The results clearly show that our proposed order, $\tau_c \rightarrow A \rightarrow \tau_d$, is superior. Placing the concise rationale $\tau_c$ first is essential. When the detailed rationale $\tau_d$ is generated first (e.g., $\tau_d \rightarrow \tau_c \rightarrow A$), the model fails to produce concise outputs at inference (106.5 tokens on ReaSeg), as it has not learned to abstract. This confirms that the predictive nature of the $\tau_c$-first ordering is the key to enabling thought compression.

**All Reward Components are Necessary.** Table 5 dissects our self-distillation reward. Removing any of the three components—the conditional indicator ($\mathbb{I}$), the similarity score ($S_{sim}$), or the conciseness score ($S_{concise}$)—leads to a noticeable degradation in either performance, efficiency, or both. For instance, removing the similarity constraint ($S_{sim}$) results in much longer rationales, as the model is not explicitly encouraged to make $\tau_c$ a faithful summary. Removing the conditional application ($\mathbb{I}$) leads to unstable training where the model might distill incorrect reasoning paths (not shown in table). This validates that our hierarchical reward formulation is critical for success.

### 4.2.2 EFFECT OF THE INFERENCE-TIME PROMPT

Table 6 explores different inference-time strategies. The baseline Seg-Zero model generates over 100 tokens. Our default WISE model, which simply omits the generation of $\tau_d$ without any special prompt, already reduces this to a much more reasonable 34-39 tokens while improving accuracy. This demonstrates the strong intrinsic brevity learned during training. Adding a brevity-focused prompt further enhances efficiency. While "*the shorter the better*" yields the absolute fewest tokens, it comes at a slight cost to performance. Our proposed WISE-S prompt, "*one sentence*", strikes the optimal balance, achieving the best overall trade-off between high cIoU and extremely low token count. This shows that a simple, targeted prompt is an effective way to fully unlock and stabilize the concise reasoning capability cultivated by our training paradigm.

Table 4: Ablation on the **generation order** of rationales and the answer. Placing the concise rationale first ($\tau_c \to A \to \tau_d$) is crucial for achieving both high accuracy and low token count at inference.

| Order | RCOg | | ReaSeg | |
|---|---|---|---|---|
| | #Tok ↓ | cIoU ↑ | #Tok ↓ | cIoU ↑ |
| $A \to \tau_c \to \tau_d$ | 74.6 | 70.1 | 75.7 | 54.4 |
| $\tau_d \to \tau_c \to A$ | 107.5 | 68.2 | 106.5 | 55.4 |
| $\tau_c \to \tau_d \to A$ | 134.7 | 70.1 | 132.7 | 55.2 |
| $\tau_c \to A \to \tau_d$ | **34.0** | **71.8** | **38.8** | **58.5** |

Table 5: Ablation of the **self-distillation reward**. All three components—the conditional indicator ($\mathbb{I}$), semantic similarity ($S_{sim}$), and conciseness score ($S_{concise}$)—are essential.

| Components | | | RCOg | | ReaSeg | |
|---|---|---|---|---|---|---|
| $\mathbb{I}$ | $S_{sim}$ | $S_{concise}$ | #Tok ↓ | cIoU ↑ | #Tok ↓ | cIoU ↑ |
| | | | 159.8 | 71.3 | 111.9 | 54.4 |
| | ✓ | ✓ | 67.8 | 70.1 | 75.2 | 54.6 |
| ✓ | | ✓ | 46.2 | 68.6 | 43.6 | 57.8 |
| ✓ | ✓ | | 78.9 | 70.7 | 74.2 | 56.3 |
| ✓ | ✓ | ✓ | **34.0** | **71.8** | **38.8** | **58.5** |

Table 6: Ablation study on the impact of **inference-time strategies**. Our default setting ($+\tau_c - \tau_d$) already provides a strong balance. The brevity-focused prompts further enhance efficiency, with WISE-S achieving the best trade-off.

| Prompt | RCOg | | ReaSeg | |
|---|---|---|---|---|
| | #Tok ↓ | cIoU ↑ | #Tok ↓ | cIoU ↑ |
| Seg-Zero (baseline) | 159.8 | 71.3 | 111.9 | 54.4 |
| WISE-F ($+\tau_c + \tau_d$) | 188.5 | 71.3 | 197.5 | 58.8 |
| Directly Remove | | | | |
| $-\tau_c \quad -\tau_d$ | 51.5 | 70.3 | 59.9 | 54.8 |
| $-\tau_c \quad +\tau_d$ | 200.0 | 68.4 | 221.4 | 53.2 |
| WISE ($+\tau_c - \tau_d$) | 34.0 | 71.8 | 38.8 | **58.5** |
| Brevity-Focused Prompt | | | | |
| *"the shorter the better"* | **15.4** | 71.6 | **15.4** | 58.2 |
| WISE-S (*"one sentence"*) | 22.9 | **72.1** | 22.7 | 58.3 |

### 4.3 ANALYSIS OF REASONING EFFICIENCY

To provide a more nuanced understanding of the efficiency gains, we visualize the distribution of reasoning token lengths on the ReasonSeg test set in Figure 4. The baseline **Seg-Zero** exhibits a wide and right-skewed distribution, indicating its reasoning process is not only long but also highly variable. In sharp contrast, our **WISE** model's distribution is tightly concentrated around a much smaller mean, a result of its learned intrinsic preference for brevity. The **WISE-S** variant further sharpens this distribution into a consistent, low-cost output, making its computational cost stable and predictable—a critical feature for real-world deployment.

This efficiency gain is not merely a reduction in length but a reflection of improved reasoning quality. As illustrated in the qualitative example in Figure 5, given a complex functional instruction, Seg-Zero engages in a convoluted, 132-token rationale and ultimately fails. Conversely, WISE-S identifies the correct object with a focused, 24-token thought. This vividly demonstrates that our thought compression encourages the model to discard irrelevant details and focus on the core logic, leading to reasoning that is not only more efficient but also more robust.

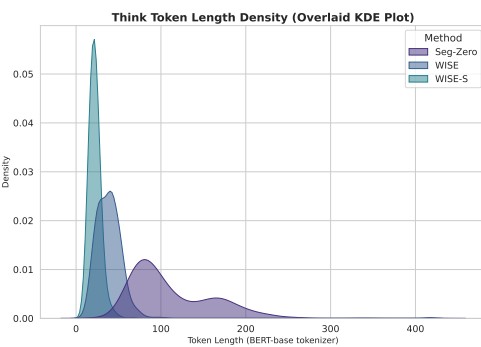

(a) Kernel Density Estimate (KDE) Plot

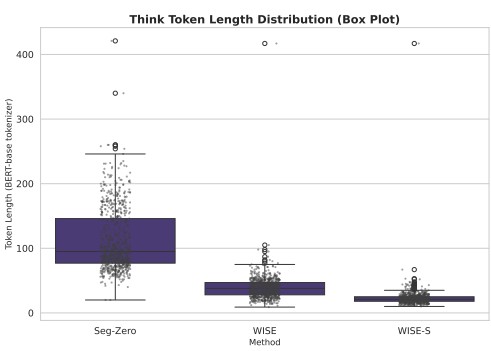

(b) Box Plot with Strip Plot Overlay

Figure 4: Distribution of the reasoning token length on the ReasonSeg test set. The KDE plot (a) shows the overall density, while the box plot (b) summarizes the statistical distribution. Both plots clearly illustrate the dramatic reduction in length and variance achieved by our WISE and WISE-S methods compared to the Seg-Zero baseline.

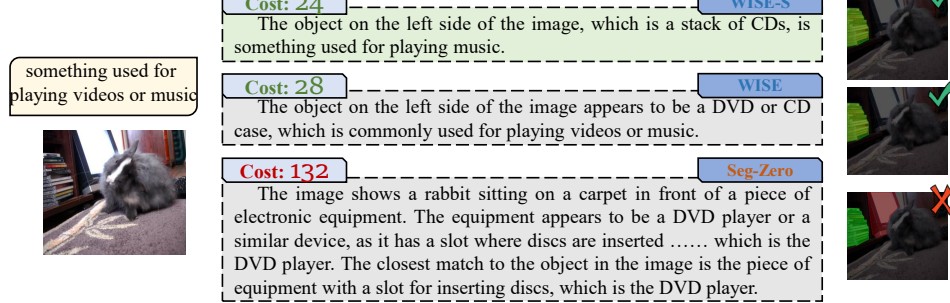

Figure 5: **Qualitative comparison on a challenging reasoning task.** Given the instruction to find something for playing media, Seg-Zero produces a long, distracted rationale and fails. In contrast, both WISE and WISE-S generate concise, correct reasoning chains, successfully identifying the stack of CDs/DVDs with a fraction of the token cost.

## 5 CONCLUSION

In this work, we addressed the critical tension between reasoning depth and computational efficiency in large multimodal models for language-guided segmentation. We introduced **WISE**, a novel training and inference framework built on the concept of **thought compression**. By restructuring the generation process to predict a concise rationale before its detailed explanation, our method fosters a genuine capability for reasoning abstraction. This is achieved through a unique, predictive training objective, guided by a conditional self-distillation reward, without relying on any human-annotated reasoning data. Our experiments demonstrate that WISE, particularly our inference-time variant WISE-S, not only achieves state-of-the-art performance on challenging benchmarks like zero-shot ReasonSeg but also drastically reduces the computational cost of reasoning by over $4\times$. This work empirically proves that deep reasoning and high efficiency are not mutually exclusive, opening the door for the deployment of powerful, yet practical, reasoning models in real-world applications.

ACKNOWLEDGMENTS

We would like to thank the LLM that served as a fantastic writing assistant throughout this project, helping to polish the language and improve the flow of our paper.

In the spirit of open research, all code and data will be released publicly after the paper is accepted.

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
