# A  QUALITATIVE ANALYSIS: DISTILLING SUCCESS AND FAILURE

To validate the robustness of our thought compression, we analyze both success and failure scenarios. Our analysis reveals that WISE-S achieves high efficiency without introducing new reasoning errors, maintaining high **Semantic Fidelity** to the original verbose reasoning.

**Success Cases: Concise Rationale as a Sufficient Summary.** As shown in Figure 1, in successful instances, WISE-S demonstrates the ability to distill the *decision logic* while discarding *visual redundancy*.

- *Attribute Grounding:* In the equestrian example (Figure 1, Bottom-Right), the detailed explanation ($\tau_d$) engages in a verbose verification process, checking the horse's position and the nature of the sport. In contrast, WISE-S ($\tau_c$) directly extracts the discriminative features—"red and white obstacle" and "foreground"—which are sufficient to localize the mask.

- *Functional Reasoning:* In the cave exploration example (Figure 1, Bottom-Left), the model must process a negative constraint ("did not consider diving"). WISE-S correctly reasons that the target area must be "above the water level," efficiently pruning the search space without the need for the extensive geological description found in the detailed chain.

These examples confirm our hypothesis that the concise rationale learns to act as a sufficient summary for the final answer, effectively bridging the gap between instruction and segmentation.

**Failure Cases: Consistent Limitations.** We closely examined cases with low IoU (Figure 2) to determine if the brevity constraint caused the failure. Interestingly, we found that **the compression mechanism is rarely the culprit**.

- **Fidelity in Failure:** In the "Warthog" example, the concise rationale ($\tau_c$) correctly identifies the target object as "tusks," fully capturing the semantic core of the verbose explanation ($\tau_d$). The failure to segment the specific tusks (likely masking the whole face) is a shared limitation in *spatial grounding* inherent to the base Vision-Language Model, occurring in both standard WISE and WISE-S modes.

- **Shared Hallucination/Ambiguity:** In the "Concept Car" example, both $\tau_c$ and $\tau_d$ exhibit circular logic (tautology), failing to identify specific visual attributes. $\tau_c$ merely summarizes the vague reasoning of $\tau_d$.

**Conclusion:** These failure cases powerfully demonstrate the effectiveness of our **Self-Distillation** objective. The model successfully internalized the reasoning—whether strong or weak—into a compressed form. The errors stem from the backbone model's capabilities, not the thought compression process itself. This confirms that WISE-S provides a "lossless" speedup in terms of reasoning quality.

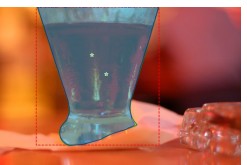

if someone in the picture wanted to enjoy a drink without having to pour it themselves, what object could they use to easily take small sips at a time?

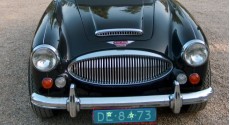

something indicating the identity of the car

$\tau_c$

The object that could be used to easily take small sips at a time is the glass in the picture.

$\tau_c$

The license plate on the car indicates the identity of the car.

$\tau_d$

The object that could be used to easily take small sips at a time is the glass in the picture. This because the glass is designed to hold a liquid and has a shape that allows for easy sipping. The presence of a handle on the glass suggests that it is meant to be held and sipped from, which is a common feature in glasses designed for drinking. The glass is filled with a red liquid, which could be a beverage, and the shape of the glass is conducive to taking small sips. The other object in the picture, which appears to be a salt shaker, is not designed for drinking and does not have a shape that would allow for easy sipping. Therefore, the glass is the most closely matched object for taking small sips at a time.

$\tau_d$

The license plate on the car is a common feature that provides information about the vehicle, including its registration and sometimes the make and model. In this image, the license plate is clearly visible and appears to be a European license plate, which suggests that the car is registered in a European country. The license plate is located at the front of the car, which is a standard position for license plates on many vehicles. The presence of the license plate indicates that it is an important identifier for the car. Additionally, the license plate is a fixed part of the car and is not likely to be removed or altered without notice, making it a reliable indicator of the car's identity. Therefore, the license plate is the most closely matched object that indicates the identity of the car.

Seg-Zero

The image shows a close-up of a glass with a red liquid, which is likely a drink. The glass is a standard drinking glass, but the question asks for an object that would allow for easy sipping. The glass in the image is not designed for sipping, as it is a standard drinking glass. However, the question implies that there might be a different object in the image that could be used for sipping. The object to the right of the glass appears to be a small, clear, and textured object, which resembles a sipping cup or a small drinking glass. This object is designed for easy sipping, as it is smaller and more compact than the main glass.

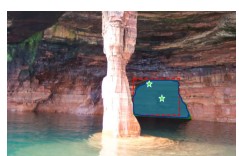

(a) **Qualitative Comparison: Efficiency vs. Distraction.** The figure demonstrates how WISE-S acts as a sufficient statistic, contrasting its focused reasoning with the baseline's verbose failure. **(Left) Functional Reasoning:** While WISE-S ($\tau_c$) efficiently identifies the glass by its affordance ("take small sips"), the baseline **Seg-Zero** (bottom) suffers from *reasoning drift*. It over-analyzes the prompt constraints, leading to a hallucinated conclusion about a non-existent "small textured object" to the right. This vivid example illustrates how thought compression can prevent the model from getting lost in irrelevant visual details. **(Right) Identity Grounding:** WISE-S correctly filters out spatial redundancy to focus on the discriminative visual attribute ("license plate") required to establish identity.

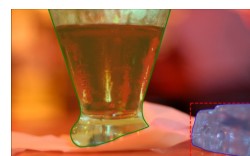

if we were at the location shown in the picture and did not consider diving underwater, what area in the picture could we explore further?

in equestrian sports, riders and horses need to overcome obstacles to complete a course. what in the picture can be used as an obstacle for the rider and horse to jump over?

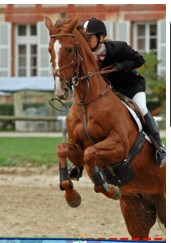

$\tau_c$

The red and white obstacle in the foreground is the obstacle that the rider and horse need to jump over.

$\tau_c$

If we were at the location shown in the picture and did not consider diving underwater, we could explore the area above the water level further.

$\tau_d$

The area above the water level in the picture appears to be a cave or a series of rock formations that are not submerged. This area is likely to offer more exploration opportunities without the need for diving. The presence of the water level suggests that the area above it is not easily accessible from the water, which means that exploring this area would require climbing or walking on the rock formations. This area could potentially reveal more about the geological structure of the location, the history of the rock formations, and the natural beauty of the area. The cave or rock formations above the water level could also provide a unique perspective on the landscape, offering a different view of the surroundings and the water below. Therefore, if we were at the location shown in the picture and did not consider diving underwater, the area above the water level could be explored further.

$\tau_d$

The red and white obstacle in the foreground is the obstacle that the rider and horse need to jump over. This is evident from the position of the horse and rider, who are in mid-air, indicating that they are in the process of jumping over an obstacle. The obstacle is located in the foreground, which is a common placement for obstacles in equestrian sports to ensure that the rider and horse have enough space to approach and clear it. The red and white color scheme is also typical for obstacles in equestrian sports, as it is easily visible and distinguishable from the surrounding environment. The horse and rider are positioned in a way that suggests they are actively jumping over this obstacle, further confirming that it is the one they are attempting to clear. Therefore, the red and white obstacle in the foreground is the most closely matched object for the rider and horse to jump over.

(b) Visual Attribute Grounding Case

Figure 1: **Success Case Study Examples.** The figures demonstrate how WISE-S acts as a sufficient statistic. (a) Shows the model efficiently identifying an object by its function ("take small sips") without verbose description. (b) Shows the model correctly filtering out spatial redundancy to focus on discriminative visual attributes ("red and white obstacle").

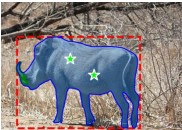

in the wild, animals often exhibit specific behaviors and adaptations that help them survive in their natural habitats. what part of the animal in the picture can be used for defense and aggression?

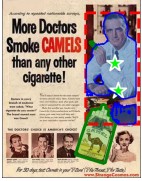

due to the harmful effects of smoking on health, many countries have implemented regulations to discourage smoking. what depicted in the picture is likely to have negative implications regarding smoking?

τ_c

The animal in the picture is a warthog, and its tusks can be used for defense and aggression.

τ_c

The image depicts an advertisement for Camel cigarettes, which is likely to have negative implications regarding smoking due to the harmful effects of smoking on health.

τ_d

The animal in the picture is a warthog, which is known for its tusks. Tusks are elongated incisor teeth that protrude from the upper jaw and are used for various purposes, including defense and aggression. In the image, the warthog has prominent tusks that are clearly visible. These tusks are a distinctive feature of warthogs and are used for a variety of behaviors, such as digging for food, defending against predators, and establishing dominance within a group. The tusks are also used in combat with other warthogs, which is a common behavior in the wild. The warthog in the picture appears to be in a natural habitat, which further supports the idea that the tusks are used for defense and aggression in the wild. The presence of the tusks in the image is a strong indicator that they are the part of the animal that can be used for defense and aggression.

τ_d

The image is an advertisement for Camel cigarettes, which is a brand of tobacco products. The presence of a doctor smoking a cigarette in the image is a significant red flag regarding the negative implications of smoking. Tobacco advertising, especially featuring healthcare professionals, has historically been used to normalize smoking and create a false sense of trust and safety around the product. This type of advertising can have negative implications for public health by potentially encouraging people, especially those who look up to healthcare professionals, to start smoking or continue smoking.\n\nFurthermore, the image includes text that states, \"More Doctors Smoke CAMELS than any other cigarette!\" This statement is likely to be misleading and could be interpreted as a endorsement of the product by doctors, which is not the case. Doctors are professionals who are well-informed about the health risks associated with smoking. By associating smoking with doctors, the advertisement is attempting to create a false narrative that smoking is a safe or healthy choice, which is a significant concern for public health.\n\nThe image also includes a call to action, \"For 30 days, test Camels in your 'T-Zone' (T for Throat, T for Taste).\" This suggests that the advertisement is trying to encourage people to try smoking, which is a harmful behavior that can lead to various health issues, including lung cancer, heart disease, and respiratory problems.\n\nIn conclusion, the image depicts an advertisement for Camel cigarettes, which is likely to have negative implications regarding smoking due to the harmful effects of smoking on health. The presence of a doctor smoking, the misleading statement about doctors smoking more Camel cigarettes than any other cigarette, and the call to action to try smoking all contribute to the negative implications of the advertisement for public health.

(a) Grounding Limitation Case

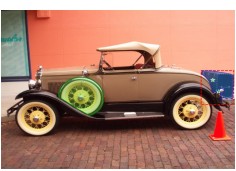

the spare wheel

in an automobile exhibition, car manufacturers often showcase their latest designs and prototypes. what in the picture can be used to showcase a new and innovative car design?

τ_c

The spare wheel is located in the trunk of the car.

τ_c

The car in the picture can be used to showcase a new and innovative car design

τ_d

The spare wheel is typically located in the trunk of a car, and in this image, the trunk is open, revealing a spare wheel. The spare wheel is positioned in the back of the car, which is a common location for this type of wheel. The open trunk provides a clear view of the spare wheel, confirming its presence. Additionally, the spare wheel is distinct from the other wheels, as it is larger and has a different design, which is typical for a spare wheel. The open trunk also allows for a clear view of the interior of the trunk, further confirming the presence of the spare wheel. Therefore, the spare wheel is the most closely matched object to the description \"the spare wheel.\"

τ_d

The car in the picture is a concept car, which is specifically designed to showcase new and innovative car designs. Concept cars are often displayed in automobile exhibitions to attract attention and highlight the latest technological advancements and design concepts. The car in the picture has a sleek, futuristic design with a low profile, large wheels, and a unique body shape, all of which are indicative of a new and innovative design. The setting, with the car on display in a showroom, further suggests that it is being showcased to the public, which is a common practice in automobile exhibitions. The presence of a screen in the background with the text \"CODE 30R\" also suggests that the car is part of a larger exhibition or event, where multiple concept cars are being displayed and showcased. Therefore, the car in the picture can be used to showcase a new and innovative car design.

(b) Circular Logic Case

Figure 2: **Analysis of Failure Cases: Semantic Fidelity amidst Grounding Errors.** These examples illustrate instances where the model fails to produce an accurate mask (Low IoU). Crucially, however, the Concise Rationale ($\tau_c$) remains a **faithful summary** of the Detailed Explanation ($\tau_d$). (a) In the top example, both $\tau_c$ and $\tau_d$ correctly identify the semantic topic but struggle to ground the abstract concept to specific pixels. (b) In the bottom example, both rationales exhibit circular logic without identifying distinct visual attributes. This indicates that the failures stem from the **underlying limitations** of the base model's spatial grounding capabilities or reasoning loops, rather than information loss caused by the WISE-S compression mechanism.