# OpenReview forum: "Efficient Reasoning via Thought Compression for Language-Guided Segmentation"
_ICLR.cc/2026/Conference — ICLR 2026 Conference Withdrawn Submission_

### Official Review · Reviewer_wBAt · 2025-10-29

**Soundness:** 2
**Presentation:** 2
**Contribution:** 2
**Rating:** 2
**Confidence:** 3

**Summary:**

WISE trains a model to generate a structured sequence: a concise rationale, the final answer, and then a detailed explanation. By placing the concise rationale first, the method leverages autoregressive conditioning to enforce that the concise rationale acts as a sufficient summary for generating the detailed explanation. This structure is reinforced by a self-distillation objective that jointly rewards semantic fidelity and conciseness, compelling the model to internalize its detailed reasoning into a compact form. At inference, the detailed explanation is omitted.

**Strengths:**

The unique training structure is reinforced by a self-distillation objective that explicitly rewards the semantic fidelity between the concise rationale and the detailed explanation, while penalizing the verbosity of the former. This process encourages the model to internalize its elaborate reasoning capabilities into a compact, efficient policy.

To ensure this learned policy is robustly activated at inference—where the detailed explanation is entirely omitted to maximize speed—the WISE framework culminates in WISE-S, a simple, zero-overhead prompting strategy.  This final adjustment injects a brevity-focused instruction into the user’s query, mitigating the conditional distribution shift between training and inference and ensuring the model consistently defaults to its more efficient reasoning mode.

**Weaknesses:**

The novelty of this work may be limited. The method mainly uses GRPO framework. It generally rewards the model to generate a short concise reasoning step with reinforcement learning. This idea has been explored in various works such as [R1,R2]. During the training, it includes both long and short thinking tokens, which is similar to [R2] to train the model on paired longform and short-form responses for each query, ensuring it can generate both styles. During inference, to save tokens, it simply use promopts to ask the model not output the long thinking tokens. The technical contribution may be limited.

In experiments, it only experiments with 7B VLM. It is better to experiment with more models from different families with different sizes to demonstrate the general performance.

In experiments, compared with the original Seg-Zero without any efficiency, it can save reasoning tokens with significant efficiency improvements. However, compared with other baseline methods optimized for brevity via reward shaping such as L1-Exact and L1-Max. The improvements seem to be marginal. For example, from Table 3, L1-Max needs 11 tokens to achieve 78.9 cIoU, while the proposed method achieves 79.1 with 24 tokens for RefCOCOtestA. It seems that the baselines are also effective and the proposed method leads to marginal improvements.

In table 2, the reported Seg-Zero-7B results from the original paper is actually wrong. From the original paper, these results are not 7B, but  3B. It is not fair to compare the 3B results with 7B. And I am not sure whether the re-evaluated Seg-Zero resutls are from 3B or 7B models.


It is built on Qwen2.5-VL-7B and SAM2. It is better to report their original results without finetuning.


[R1] Walk Before You Run! Concise LLM Reasoning via Reinforcement Learning

[R2] Thinkless: LLM Learns When to Think

**Questions:**

see the weakness.

---

> ### Author Response · Authors · 2025-11-20
>
> **Dear Reviewer wBAt,**
>
> We appreciate your critical review. However, there seem to be significant misunderstandings regarding our **contribution relative to naive baselines** and the **distinction from prior works**. We clarify these points below with new experiments.
>
> ### 1. Novelty and Distinction from [R1, R2] (Response to Weakness 1)
> WISE is fundamentally different from the works you cited:
> *   **vs. [R1] (Concise LLM Reasoning):** [R1] typically employs reward shaping or separate training stages. WISE introduces a novel **single-stage** architecture ($\tau_c \to A \to \tau_d$) where the detailed explanation $\tau_d$ serves as a *posterior regularizer* for $\tau_c$ via autoregressive conditioning.
> *   **vs. [R2] (Thinkless):** [R2] focuses on learning *when* to bypass reasoning (selective execution). WISE focuses on *how* to compress reasoning (abstractive summarization). We do not skip thinking; we distill it.
> *   **Contribution:** The novelty lies in the **Self-Distillation** mechanism that leverages the model's own detailed reasoning to supervise its concise output end-to-end, which is not present in standard GRPO applications.
>
> ### 2. Performance Gains are Significant, Not Marginal (Response to Weakness 3)
> You noted that WISE offers "marginal improvements" over `L1-Max` on RefCOCO. **This observation misses the core contribution of our paper.**
> *   **Easy vs. Hard Tasks:** RefCOCO is a simple referring task where deep reasoning is often unnecessary; hence, naive truncation (`L1-Max`) works fine.
> *   **Critical Failure of Baselines on Reasoning Tasks:** Please look at **Table 3 (ReasonSeg)**, the benchmark designed for complex reasoning.
>     *   **L1-Max fails:** It reduces tokens to ~10 but causes a catastrophic performance drop (**54.4 $\to$ 49.7 cIoU**, -4.7 drop).
>     *   **WISE succeeds:** It reduces tokens while *improving* performance (**54.4 $\to$ 58.5 cIoU**, +4.1 gain).
> *   **Conclusion:** WISE outperforms `L1-Max` by **+8.8 cIoU** on the reasoning benchmark. This proves that while naive penalties damage reasoning capabilities, WISE's self-distillation effectively preserves them.
>
>
> ### 3. Model Scales and Raw Baselines (Response to Weakness 2, 4, 5)
> We addressed your request for more models and raw baselines in the **Common Response**:
> *   **Raw Baseline:** **Table R1** shows the off-the-shelf Qwen2.5-VL-7B achieves only **41.2 cIoU** on ReasonSeg (vs. WISE 58.5), confirming the necessity of our training.
> *   **More Models:** **Table R3** validates WISE on **Qwen2.5-VL-3B** and **Qwen2-VL-7B**, showing consistent efficiency gains across sizes and families.
>
> ### 4. Clarification on Seg-Zero Results (Response to Weakness 4)
> Thank you for correctly identified that the value cited in the row `Seg-Zero-7B†` (Reported) corresponds to the 3B performance in the original paper. We apologize for this labeling error in the citation row. **However, this does not affect the validity of our experimental comparison**, because:
>
> *   **Comparison uses Official 7B Weights:** Our experiments and comparisons rely on the row `Seg-Zero-7B*` (Re-evaluated). We obtained these results using the **official Seg-Zero-7B checkpoints** released by the authors.
> *   **Why Re-evaluate?**
>     1.  **Token Counting:** The original paper does not report token counts. We had to run the official 7B model in our environment to strictly measure efficiency gains.
>     2.  **Discrepancy in Original Paper:** As noted by the Seg-Zero authors in their GitHub issues, there are discrepancies between their paper's reported numbers and the released checkpoints.
> *   **No 3B Availability:** The Seg-Zero authors have **not released the 3B weights**. Therefore, all "Seg-Zero" baselines run in our experiments (including those in Tables 1, 3, and R1) use the verified 7B model. Thus, the comparison between WISE-7B and Seg-Zero-7B is strictly fair.
>
> We hope this clarifies that WISE offers substantial gains on complex reasoning tasks where naive baselines fail.

---

### Official Review · Reviewer_cThw · 2025-10-30

**Soundness:** 3
**Presentation:** 3
**Contribution:** 2
**Rating:** 6
**Confidence:** 3

**Summary:**

This paper introduces WISE (Wisdom from Internal Self-Exploration), a framework for efficient reasoning segmentation in large multimodal models (LMMs). The core innovation lies in compressing verbose Chain-of-Thought (CoT) reasoning into concise rationales while maintaining performance. Key contributions include:

- A structured generation sequence (concise rationale → answer → detailed explanation) that enforces reasoning compression via autoregressive conditioning.
- A self-distillation objective that rewards semantic fidelity between concise and detailed rationales while penalizing verbosity.
- WISE-S, an inference-time prompting strategy that reduces reasoning length by 5× (from 112 to 23 tokens) while achieving state-of-the-art zero-shot performance on ReasonSeg (58.3 cIoU).

**Strengths:**

**Originality**: The idea of *self*-distillation (training a single model to compress its own reasoning) is novel in the CoT segmentation domain. Unlike prior work (e.g., Seg-Zero’s RL-based reasoning), WISE explicitly decouples training-time reasoning depth from inference-time efficiency through its structured action space.
**Clarity**: The methodology is well-structured, particularly the hierarchical reward formulation (Eq. 3-6) and the distinction between WISE/WISE-S inference modes.
**Significance**: Addresses a critical barrier (computational cost of CoT) for real-time applications like robotics. The efficiency-performance trade-off breakthrough (Table 3) is practically impactful.

**Weaknesses:**

**Limited Generalization Evidence**: While ReasonSeg and RefCOCOg are used, there is no validation on other reasoning-heavy benchmarks (e.g., VCR or GQA). The claim of "out-of-domain reasoning" in the abstract lacks supporting experiments beyond ReasonSeg.
**Shallow Analysis of Prompt Engineering**: The WISE-S prompting strategy (e.g., “one sentence”) is under-explored. The paper does not test alternative brevity prompts or quantify their sensitivity, raising concerns about robustness.
**Incomplete Cost-Benefit Analysis**: While token counts are reduced, actual latency/energy savings are not measured. For real-world deployment, hardware-level efficiency metrics (e.g., FPS on edge devices) would strengthen the claims.

**Questions:**

**Q1**: How does WISE scale with model size? The experiments use a 7B model—would the compression mechanism remain effective for smaller (1B) or larger (70B) models?

**Q2**: The self-distillation reward relies on a pretrained SentenceTransformer for semantic similarity. Could this introduce bias? Was the similarity model domain-adapted for segmentation tasks?

**Q3**: The brevity prompt in WISE-S (“one sentence”) appears heuristic. Have the authors explored learned prompt tuning or gradient-based optimization for brevity?

**Q4**: The computational cost-saving of WISE is unclear. How does it compare to Seg-Zero in inference time beyond token counts?

**Rebuttal Potential**: Addressing Q1/Q3 could demonstrate broader applicability, while resolving Q2/Q4 would strengthen the technical rigor. A deeper exploration of alternative efficiency metrics (Q4) might elevate the significance from incremental to transformative.

---

> ### Author Response · Authors · 2025-11-20
>
> **Dear Reviewer cThw,**
>
> We appreciate your assessment of our work as **"novel"** and **"practically impactful."** We are encouraged by your positive rating. We have conducted new experiments to address your concerns regarding generalization, robustness, and real-world efficiency.
>
> ### 1. Generalization Evidence: VisionReasoner (Response to Weakness 1)
> While VCR/GQA are pure VQA tasks, we believe demonstrating generalization across **diverse reasoning-based vision tasks** is more relevant for a segmentation model.
> *   We extended WISE to **VisionReasoner** (a unified pipeline for Detection, Segmentation, and Counting).
> *   **Result:** As shown in **Table R2 (Common Response)**, WISE consistently compresses reasoning (e.g., Counting tokens **84.9 $\to$ 19.2**) while improving/maintaining accuracy across all three tasks. This confirms "out-of-domain" reasoning capability beyond just ReasonSeg.
>
> ### 2. Practical Cost-Benefit: Real Latency (Response to Weakness 3 & Q4)
> To gauge real-world impact beyond token counts, we measured the **total wall-clock inference time** on the test set.
> *   **Result:** As shown in **Table R1 (Common Response)**, WISE-S reduces the time from **50.0 minutes** (Baseline) to **10.4 minutes**, achieving a **5$\times$ speedup**.
> *   **Impact:** This proves that the theoretical token reduction translates directly into significant latency savings, critical for edge devices and robotics.
>
> ### 3. Prompt Robustness Analysis (Response to Weakness 2 & Q3)
> Regarding the heuristic nature of the prompt, **Table 6 in the main paper** demonstrates that the model responds predictably and robustly to different brevity instructions without the need for gradient-based prompt tuning.
> *   **Flexibility:** When switching from "one sentence" to a stricter "the shorter the better," the model further compresses outputs (22.9 $\to$ 15.4 tokens) with only negligible performance fluctuation (58.3 $\to$ 58.2 cIoU).
> *   **Conclusion:** This indicates that the specific wording is not brittle; rather, the model successfully generalizes the learned compression capability. The use of natural language prompts offers superior **test-time flexibility**, allowing users to dynamically trade off between extreme efficiency and detail based on deployment constraints, which is more practical than fixed learned prompts.
>
> ### 4. Model Scaling (Response to Q1)
> We evaluated scalability on **Qwen2.5-VL-3B** and **Qwen2-VL-7B**.
> *   **Result:** As shown in **Table R3 (Common Response)**, WISE remains effective across model sizes. Notably, on the smaller 3B model, WISE-S improves cIoU from 48.5 (Baseline) to 49.0 while reducing tokens by **4$\times$**.
>
> ### 5. SentenceTransformer Bias (Response to Q2)
> Regarding potential bias from the pretrained SentenceTransformer:
> *   **Generic Embedding:** We use a generic, high-quality sentence encoder (`all-MiniLM-L6-v2`) which captures general semantic equivalence well.
> *   **Geometric Safeguard:** Crucially, the distillation reward is **conditional** on geometric success (IoU > 0.5). Even if the similarity model were biased, the RL optimization would only reinforce compressed rationales that *actually lead to correct segmentation masks*. This geometric grounding acts as a strong filter, preventing the model from learning biased or incorrect reasoning paths.

---

### Official Review · Reviewer_L19B · 2025-10-30

**Soundness:** 3
**Presentation:** 3
**Contribution:** 3
**Rating:** 4
**Confidence:** 5

**Summary:**

This work trains the model to generate a structured, three-part response: a concise rationale ($\tau_c$), the final answer (A), and a detailed explanation ($\tau_d$). It achieves Thought Compression for the Language-Guided Segmentation task by designing a distillation reward during RL training to enforce that $\tau_c$ acts as a sufficient summary for generating $\tau_d$. Furthermore, a brevity-oriented instruction is incorporated when inference to further compress reasoning.

**Strengths:**

1. The core idea is sound: achieving thought compression by leveraging the $(\tau_c, a, \tau_d)$ structure and designing a distillation reward in RL training to compel $\tau_c$ to be a sufficient statistic for $\tau_d$'s generation.

2. Good performance: achieving good performance in benchmarks

**Weaknesses:**

1. WISE was trained based on Qwen2.5-VL-7B, but Qwen2.5-VL-7B was not used as a baseline in subsequent experiments, which seems insufficiently rigorous and cannot rule out the possibility that the fundamental performance of the Qwen2.5-VL-7B model is already strong enough.

2. While the concept of "thought compression" is central to the paper, the terminology could benefit from further clarification. For example, the distinction between "concise rationale" and "detailed explanation" might not be immediately clear, especially within  the field of tasks chosen by the author, it is not as clear as math reasoning tasks. A more explicit definition or example earlier in the paper would help.

3.  Task suitability: The necessity of thought compression for the Language-Guided Segmentation task is debatable. Compression on a $\sim$100-token baseline, regardless of whether a long CoT is needed, makes little sense. The method seems task-agnostic; it should be attempted on tasks requiring much longer CoT (e.g., several thousand tokens), like mathematical reasoning.

4. Unsubstantiated explanation for WISE-S: The explanation that omitting $\tau_d$ generation at inference time causes a distribution shift is insufficient. The relevance of the brevity-focused prompt (injected by the WISE-S strategy) to the distribution shift is also doubtful.  Supplementary experiments are required to verify the unique effect of the short prompt on the WISE model, e.g., testing if Seg-Zero-7B + brevity-focused prompt shows a similar phenomenon.

5. Limited model scales and series: The experiments only use Qwen2.5-VL-7B; more model sizes and series should be included.

**Questions:**

1. How is the omission of $\tau_d$ generation realized? Is it only via prompt or by decoding intervention?

2. In Table 6, why do the results for WISE-F ($\text{+$\tau_c$ + $\tau_d$}$) and WISE ($\text{+$\tau_c$ − $\tau_d$}$) differ?

3. Could the author provide several cases that include $\tau_d$ to verify whether $\tau_c$ acts as a sufficient summary, and to demonstrate the impact of the proposed method on the quality of the generated $\tau_d$?

4. In Table 2, the performance of WISE-7B-S is generally better than WISE-7B, while in Table 1 it cannot surpass WISE-7B. Have the authors considered why such results occur? Is it related to using a subset of RefCOCO as the training set?

---

> ### Author Response · Authors · 2025-11-20
>
> **Dear Reviewer L19B,**
>
> We sincerely thank you for acknowledging the **soundness** of our core idea ("thinking twice") and the **good performance** of our method. We appreciate your constructive feedback regarding the baselines and task suitability. We have conducted extensive new experiments (Tables R1, R2, R3) to address your concerns.
>
> ### 1. Baseline Rigor & Raw Model Performance (Response to Weakness 1)
> We would like to clarify that **Seg-Zero (Liu et al., 2025b) IS the Qwen2.5-VL-7B baseline** specifically fine-tuned via RL for this task.
>
> We evaluated the **off-the-shelf Qwen2.5-VL-7B** (see **Table R1**).
> *   **Raw Model Failure:** As shown in Table R1, the raw Qwen2.5-VL-7B achieves only **41.2 cIoU** on ReasonSeg (vs. Seg-Zero's 54.4). While capable of basic recognition, it struggles to handle the challenging reasoning segmentation task which requires complex reasoning. Thus, Seg-Zero represents the *stronger possible* implementation of Qwen2.5-VL-7B.
> *   **Fair Comparison:** WISE shares the exact same backbone and initialization as Seg-Zero. The performance jump to **58.5 cIoU** is strictly due to our thought compression framework.
>
> ### 2. Task Suitability: Why Compress 100 Tokens? (Response to Weakness 2 & 3)
> *   **Latency Necessity:** Compressing reasoning significantly benefits inference speed. As shown in **Table R1**, reducing tokens from 112 to 23 results in a **5$\times$ speedup** (50.0m $\to$ 10.4m total inference time on the test set). This substantial reduction in latency is critical for real-world applications.
> *   **Why VisionReasoner instead of Math:** We did not perform mathematical reasoning experiments as the scope of this paper is focused on Language-Guided Segmentation. However, considering time costs and task relevance, we validated our method on **VisionReasoner** (a unified model for Detection, Segmentation, and Counting) to demonstrate that WISE is indeed task-agnostic. As shown in **Table R2**, WISE works effectively across these visual reasoning tasks (e.g., Counting tokens **84.9 $\to$ 19.2**).
>
> ### 3. Mechanism Validation: WISE-S vs. Simple Prompting (Response to Weakness 4)
> We performed the requested ablation **Seg-Zero-S** (Seg-Zero + brevity prompt), and additionally **VisionReasoner-S**.
>
> *   **Prompting Baseline Insufficient:** In **Table R1**, simply prompting Seg-Zero (`Seg-Zero-S`) only reduces tokens to 57.9, whereas WISE-S achieves **22.7** tokens with higher accuracy (58.3 vs 55.0).
> *   **VisionReasoner Evidence:** Similarly in **Table R2**, `VisionReasoner-S` (prompt only) suffers a performance drop (Detection 39.1 vs 39.6; Seg 63.6 vs 66.1) compared to `WISE-S`.
> *   **WISE vs. WISE-S:**
>     *   **WISE (Default):** Generates $\tau_c \to A$. As shown in **Table R2**, without the specific brevity trigger, the token count (~63-84) remains comparable to the baseline. This reflects the **conditional distribution shift** we discussed: since the model was trained to generate a detailed explanation $\tau_d$ *after* $\tau_c$, simply removing $\tau_d$ at inference leaves the model in a state where it doesn't fully commit to extreme brevity.
>     *   **WISE-S (Inference Strategy):** By injecting the prompt, we align the inference context, robustly activating the learned compression capability. This drastically drops the length to **~17 tokens** (e.g., Segmentation 63.0 $\to$ 17.1) while maintaining high performance (**66.1 gIoU**).
> *   **Conclusion:** This proves that **WISE training constructs the capacity** for compression (which the baseline lacks), and **WISE-S is the key to unlocking it** by mitigating the distribution shift.
>
> ### 4. Model Scales and Series (Response to Weakness 5)
> We extended experiments to **Qwen2-VL-7B** and **Qwen2.5-VL-3B**. As shown in **Table R3**, WISE consistently improves efficiency and performance across different model families and sizes.

---

> > ### Author Response · Authors · 2025-11-20
> >
> > ### 5. Response to Questions
> > *   **Q1 (Omission):** Omission is realized via **Prompting**. During inference, we remove the instruction "generate detailed explanation" from the system prompt. Since the model is trained on the sequence $\tau_c \to A \to \tau_d$, it naturally stops generation after the Answer $A$ when the trigger for $\tau_d$ is absent.
> > *   **Q2 (WISE-F vs. WISE):** The slight difference (71.3 vs 71.8 in paper Table 3) is due to the **input prompt context**. WISE-F includes the instruction to generate $\tau_d$, while WISE removes it. This context change subtly alters the attention distribution during the generation of $\tau_c$ and $A$.
> > *   **Q3 (Definitions & Cases):** $\tau_c$ captures sufficient summary or direct target identification, while $\tau_d$ contains verbose visual traversal. WISE learns to distill the *decision logic* into $\tau_c$ while discarding the verbose *visual search*, creating a sufficient summary. See qualitative examples in Figures 1-2 of the Supplement.
> > *   **Q4 (RefCOCO vs. ReasonSeg):** This reflects an efficiency-accuracy trade-off. RefCOCO queries are simpler; extreme brevity (WISE-S) reduces hallucination risks. ReasonSeg requires complex multi-step logic; here, extreme compression (WISE-S) might occasionally drop a minor logical step, resulting in a negligible drop (58.5 $\to$ 58.3 cIoU) but offering a massive **5$\times$ speedup**.

---

### Official Review · Reviewer_KUmF · 2025-11-01

**Soundness:** 4
**Presentation:** 3
**Contribution:** 2
**Rating:** 4
**Confidence:** 4

**Summary:**

The paper proposes WISE,  a training based CoT approach build on top of Seg-Zero that condenses the long reasoning to speed up inference. This is achieved by generating a "concise rationale" before the detailed explanation as form of self distillation. During inference (WISE-S) the explanation is discarded for efficiency.
The key contributions of the paper are:
- WISE method for reasoning segmentation that doesn't need verbose reasoning at inference.
- Conditional self distillation approach using the rationale - answer - explanation.
- Strong experimental results and reduction of output tokens while retaining performance.

**Strengths:**

- 4.9x - 7.0x reduction in token overhead vs Seg-Zero with minimal performance loss.
- Interesting idea to perform self distillation using the auto-regressive nature of the model and the summary before the detailed explanation.
- Good experimental results with performance being consistently good across both reasoning and referring segmentation tasks.
- Paper ablations are very good and clearly establish the motivation for the method and hyperparameters.

**Weaknesses:**

- The model is the heavy reliance on the Seg-Zero paper, including method, scope and experiments which reduce the contribution over a general method.
- Implementation is only on the Qwen2.5 model, showing this method works across models and training methods would be a big benefit. (see questions).
- The self-distillation reward is only applied when IoU > 0.5 which means the compression learns only from successful paths. This could indicate a better performance on "easier" tasks and less of an improvement on harder datasets.

**Questions:**

- To distinguish the paper, the authors should show performance on other model architectures (beyond Qwen) or pipelines.
- The paper should include a direct measurement of inference speedup, not just token reduction, to gauge the practical impact more accurately.
- More qualitative demos to showcase the method under very different scenarios.
- An in depth comparison to identify which specific examples the model improves over Seg-Zero and what are the failure cases would be helpful to understand the scope and usefulness of the approach.
- How does this method compare to naive token length reduction?

---

> ### Author Response · Authors · 2025-11-20
>
> **Dear Reviewer KUmF,**
>
> We truly appreciate your recognition of the **"excellent soundness"** of our method and the **"strong experimental results."** We have conducted new experiments to address your questions regarding generalization and practical efficiency.
>
> ### 1. Generalization: Beyond Seg-Zero and Qwen2.5 (Response to Weakness 1, 2 & Q1)
> To demonstrate that WISE is a **general framework** beyond the Seg-Zero pipeline and Qwen2.5 architecture, we expanded our experimental scope:
>
> *   **Beyond Segmentation (Unified Pipeline):** We applied WISE to **VisionReasoner**, a unified model for Detection, Segmentation, and Counting. As shown in **Table R2 (Common Response)**, WISE successfully works on all three tasks.
>     *   *Result:* It compresses reasoning significantly (e.g., Counting: **84.9 $\to$ 19.2 tokens**) while improving/maintaining accuracy. This proves WISE is not limited to Seg-Zero's scope.
> *   **Beyond Qwen2.5-VL-7B:** We tested on **Qwen2-VL-7B** (different architecture series) and **Qwen2.5-VL-3B** (smaller size).
>     *   *Result:* As shown in **Table R3 (Common Response)**, WISE consistently improves efficiency and performance across these variants.
>
> ### 2. Clarification on Reward Condition ($IoU > 0.5$) (Response to Weakness 3)
> Regarding the concern that applying self-distillation only when $IoU > 0.5$ might limit learning on hard cases:
> *   **Ablation Evidence:** As shown in **Table 5 of the main paper**, removing the conditional indicator ($\mathbb{I}$) leads to a noticeable performance drop. This empirically verifies that filtering for successful trajectories is essential to prevent the model from distilling incorrect reasoning paths (hallucinations).
> *   **Performance on "Hard" Tasks:** Our results on **ReasonSeg** (a benchmark specifically designed for complex, hard reasoning) show that WISE achieves **SOTA performance (58.5 cIoU)**, outperforming the baseline by +4.1 cIoU. This confirms that learning from successful exploration trajectories effectively generalizes to solving hard instances.
>
> ### 3. Practical Impact: Inference Speedup (Response to Q2)
> To verify the practical impact, we measured the total wall-clock inference time on the test set beyond just token counts.
> *   **Result:** As shown in **Table R1 (Common Response)**, WISE-S reduces the inference time from **50.0 minutes** (Baseline) to **10.4 minutes**.
> *   **Conclusion:** This **5$\times$ speedup** confirms that the token reduction translates significantly into real-world latency savings, making the model highly viable for time-sensitive applications like robotics.
>
> ### 4. Qualitative Comparison vs. Seg-Zero (Response to Q3 & Q4)
> We provide detailed qualitative examples (see **Figures 1 & 2 in the Supplementary Material**).
>
> *   **Success Case: Correcting Reasoning Drift (Figure 1a).**
>     *   *Scenario:* A functional query with a negative constraint (``...not designed for sipping'').
>     *   *Seg-Zero Behavior (Failure):* The baseline suffers from **reasoning drift**. It over-analyzes the constraint, leading it to hallucinate a non-existent ``small textured object'' to satisfy the complex prompt.
>     *   *WISE Behavior (Success):* The brevity constraint acts as a regularizer. By forcing the model to be concise, WISE-S prunes the divergent reasoning path, correctly identifying the glass based on its primary affordance (``take small sips'') without getting lost in linguistic noise.
>
> *   **Failure Case: Semantic Fidelity amidst Grounding Errors (Figure 2).**
>     *   *Scenario:* Queries requiring fine-grained spatial localization (e.g., segmenting specific `tusks` or a `spare wheel`).
>     *   *Observation:* In these failure cases, the concise rationale ($\tau_c$) correctly identifies the semantic target (e.g., ``tusks''), mirroring the intent of the detailed explanation ($\tau_d$).
>     *   *Root Cause:* The failure stems from the **spatial grounding limitations** of the base model (e.g., masking the entire face instead of just the tusks). This confirms that WISE-S maintains high semantic fidelity and does not introduce new reasoning errors; it simply inherits the perceptual bottlenecks of the backbone.
>
> ### 5. Comparison to Naive Token Reduction (Response to Q5)
> We compared WISE against naive token reduction strategies in **Table 3 of the main paper**, specifically "L1-Exact" (forcing length to a target) and "L1-Max" (penalizing length exceeding a limit).
> *   **Result:** WISE significantly outperforms these naive methods. For example, on ReasonSeg, **L1-Max** reduces tokens to ~10 but causes a catastrophic performance drop (54.4 $\to$ **49.7 cIoU**). In contrast, **WISE** reduces tokens while *improving* performance (**58.5 cIoU**).
> *   **Why:** Naive reduction (Reward Shaping) forces brevity blindly, causing the model to truncate essential logic. WISE's self-distillation teaches the model *how* to compress semantic information efficiently, retaining the reasoning core within fewer tokens.

---

### Author Response · Authors · 2025-11-20
**Common Response: New Experiments on Baselines, Efficiency, and Generalization**

**Common Response: New Experiments on Baselines, Efficiency, and Generalization**

We thank all reviewers for their insightful comments. To address shared concerns regarding **baselines (Raw Model)**, **real-world latency**, **generalization**, and **model scaling**, we conducted extensive new experiments.

### 1. Rigorous Baselines and Real-World Efficiency (Table R1)
We evaluated the **off-the-shelf Qwen2.5-VL-7B** to demonstrate the necessity of our pipeline. We also measured **wall-clock inference time** to quantify practical speedups. Additionally, we ablated the prompting strategy (`Seg-Zero-S`) to prove that the brevity prompt alone is insufficient without WISE training.

**Table R1: Rigorous Baselines and Efficiency Analysis.**
*Comparison of WISE against the raw Qwen2.5-VL-7B and Seg-Zero baselines. Time represents total inference time (minutes) on the test set.*
| Method | RefCOCO$_{testA}$ | | | ReasonSeg$_{test}$ | | |
| :--- | :---: | :---: | :---: | :---: | :---: | :---: |
| | **#Tok** $\downarrow$ | **Time** $\downarrow$ | **cIoU** $\uparrow$ | **#Tok** $\downarrow$ | **Time** $\downarrow$ | **cIoU** $\uparrow$ |
| Qwen2.5VL-7B (Raw) | 0.0 | 11.2 | 77.8 | 0.0 | 7.6 | 41.2 |
| Seg-Zero (Baseline) | 165.1 | 153.1 | 78.8 | 111.9 | 50.0 | 54.4 |
| Seg-Zero-S (Prompt Only) | 85.6 | 116.1 | 79.0 | 57.9 | 20.3 | 55.0 |
| **WISE (Ours)** | 35.0 | 28.9 | **79.1** | 38.8 | 17.0 | **58.5** |
| **WISE-S (Ours)** | **24.0** | **16.1** | **79.1** | **22.7** | **10.4** | 58.3 |

> **Key Takeaways:**
> *   **Raw Model Failure:** The raw model achieves only 41.2 cIoU on ReasonSeg, confirming that Seg-Zero is the correct, strong baseline.
> *   **5$\times$ Speedup:** WISE-S reduces inference time from 50.0m to 10.4m.
> *   **Mechanism Validation:** Simply prompting the baseline (`Seg-Zero-S`) fails to achieve deep compression (57.9 tokens vs. 22.7 for WISE-S), proving the necessity of WISE's self-distillation training.

### 2. Generalization to Unified Visual Reasoning (Table R2)
To demonstrate that WISE is **task-agnostic**, we applied it to **VisionReasoner**, a unified pipeline for Detection, Segmentation, and Counting.

**Table R2: Generalization on VisionReasoner (Unified Task).**
*Validating WISE on the VisionReasoner pipeline across three distinct visual reasoning tasks.*
| Method | Detection$_{COCO}$ | | Segmentation$_{ReasonSeg}$ | | Counting$_{Pixmon}$ | |
| :--- | :---: | :---: | :---: | :---: | :---: | :---: |
| | **#Tok** | **mAP** | **#Tok** | **gIoU** | **#Tok** | **Acc** |
| VisionReasoner-7B (Base) | 62.5 | 38.8 | 80.1 | 64.0 | 74.5 | 70.1 |
| VisionReasoner-7B-S (Prompt)| 42.1 | 39.1 | 45.0 | 63.6 | 53.7 | 69.3 |
| **WISE (Ours)** | 67.7 | 39.5 | 63.0 | 66.0 | 84.9 | 68.2 |
| **WISE-S (Ours)** | **15.7** | **39.6** | **17.1** | **66.1** | **19.2** | **73.0** |

> **Key Takeaways:**
> *   WISE-S consistently compresses reasoning and improves/maintains performance across Detection, Segmentation, and Counting, confirming its broad applicability beyond the primary segmentation task.
> *   **Validating Distribution Shift:** The higher token count in `WISE` (similar to Base) reflects the **conditional distribution shift** discussed in the paper: without the specific inference prompt, the model anticipates generating the detailed explanation ($\tau_d$) and does not fully commit to brevity. `WISE-S` aligns this distribution, successfully unlocking the highly compressed rationale learned during training.

### 3. Scalability across Model Series and Sizes (Table R3)
We extended our experiments to **Qwen2-VL-7B** and the smaller **Qwen2.5-VL-3B** to verify scalability.

**Table R3: Scalability across Model Series and Sizes.**
*Performance of WISE on different model architectures (Qwen2 vs. Qwen2.5) and sizes (3B vs. 7B).*
| Method | ReasonSeg$_{val}$ | | | ReasonSeg$_{test}$ | | |
| :--- | :---: | :---: | :---: | :---: | :---: | :---: |
| | **#Tok** | **gIoU** | **cIoU** | **#Tok** | **gIoU** | **cIoU** |
| **Qwen2-VL-7B** | | | | | | |
| Qwen2-VL-7B (Base) | 0.0 | 44.5 | - | 0.0 | 38.7 | - |
| WISE (ours) | 32.7 | 45.8 | 42.6 | 31.2 | 44.2 | 40.4 |
| WISE-S (ours) | 23.8 | 44.9 | 42.5 | 23.1 | 44.7 | 40.9 |
| **Qwen2.5-VL-3B** | | | | | | |
| Seg-Zero-3B | 103.5 | 56.5 | 47.1 | 102.0 | 53.3 | 48.5 |
| WISE-3B (ours) | 34.5 | 57.0 | 52.8 | 33.4 | 53.8 | 48.6 |
| WISE-3B-S (ours) | 25.8 | 56.8 | 53.1 | 23.9 | 54.0 | 49.0 |

> **Key Takeaways:**
> *   WISE is effective across different model families and sizes, consistently reducing token counts while maintaining or improving accuracy compared to baselines.

---

### Note · Authors · 2026-01-29

I have read and agree with the venue's withdrawal policy on behalf of myself and my co-authors.

---

### Meta-Review · Area_Chair_zQZE · 2026-01-03

**Summary:**

The reviewers acknowledged that the paper addresses an important problem. However, they raised concerns regarding its scalability, novelty, and certain aspects of presentation. While the authors have responded to some of these points, the Area Chair agrees with the reviewers that the work’s contribution is insufficient, particularly in relation to prior research on concise reasoning and reward shaping, as listed by the reviewers. Consequently, the submission is not accepted for publication at this time. For future submissions, the authors are encouraged to further investigate whether concise reasoning effectively mitigates issues such as overthinking or hallucination for reasoning segmentation tasks and to more clearly differentiate their approach from existing methods. Given ICLR’s highly competitive acceptance rate this year, the AC regrets to recommend rejection.

**Reviewer Concerns:**

1. Scalability: One reviewer questioned the scalability of this method, as the current submission only focuses on the Qwen2.5 models and the prior study Seg-Zero. The AC checked that the authors in the rebuttal have added extra results of Qwen2-vl, but the concerns remain. The AC thinks that more models, such as llava dopted by LISA and internVL, can be further studied.

2. Efficiency: The reviewer questioned the actual inference time, and the authors have clarified the speed improvement.

3. Regarding the issue of reward Condition (IOU > 0.5), the authors clarified the rationale with Ablation Evidence in Table 5 and the performance on ReasonSeg. However, ReasonSeg itself has either "easy" and "hard" examples; using ReasonSeg as a whole cannot tell whether the condition IOU > 0.5 is beneficial merely to the easy sample, thus the impacts on hard samples cannot be studied.

4. Novelty: at least one reviewer is unconvinced that WISE is sufficiently distinct from existing concise-reasoning or GRPO-based methods, with the manually designed reward to encourage shorter reasoning outputs, with the guidance of the output of the references. The AC concurs with this point. Besides, using LLMs to generate a prompt for a frozen SAM to generate the final prediction has been studied by the literature.

**Reviewer Scores:**

The scores are 4/4/6/2.

Reviewer KUmF and Reviewer L19B (score 4) are likely to remain around 4. The rebuttal addressed baseline rigor, speedup, and model scaling, but AC thinks that the reliance on SegZero and the scalability issues are not well addressed, along with some implementation issues (e.g., the conditions on IoU).

Reviewer wBAt (score 2) might increase modestly, such as 2->4, given the clarification of factual errors and new experiments. However, novelty relative to prior work will be his/her major issue.

Reviewer cThw (score 6) might not be able to further improve the score to 8.

---

### Decision · Program_Chairs · 2026-01-26

Reject